# The Role of Chronic Fatigue in Patients with Crohn’s Disease

**DOI:** 10.3390/life13081692

**Published:** 2023-08-05

**Authors:** Marcin Włodarczyk, Adam Makaro, Mateusz Prusisz, Jakub Włodarczyk, Marta Nowocień, Kasper Maryńczak, Jakub Fichna, Łukasz Dziki

**Affiliations:** 1Department of General and Oncological, Medical University of Lodz, Pomorska 251, PL 90-213 Lodz, Poland; 2Department of Biochemistry, Medical University of Lodz, Mazowiecka 5, PL 92-215 Lodz, Poland

**Keywords:** Crohn’s disease, fatigue, pathophysiology, chronic fatigue syndrome

## Abstract

Crohn’s disease (CD) is a chronic, relapsing disorder belonging to inflammatory bowel diseases (IBD). It is manifested by relapsing transmural inflammation found in any segment of the gastrointestinal tract. Chronic fatigue is a common and underrecognized symptom of CD for which the prevalence is much higher in the population of CD patients compared to the healthy population. It stems from an intricate web of interactions between various risk factors, and its pathophysiology is still not fully understood. The implementation of routine screening and a holistic, multidisciplinary approach involving psychological support may be crucial in the management of CD patients with chronic fatigue. There is currently no single intervention aimed at decreasing fatigue alone, and its treatment is especially difficult in patients with fatigue persisting despite clinical and endoscopic remission. Extensive research is still needed in order to be able to predict, prevent, identify, and ultimately treat fatigue associated with CD. The aim of this review is to summarize the knowledge on the etiology, diagnosis, and treatment of chronic fatigue in CD patients.

## 1. Introduction

Crohn’s disease (CD) is a chronic, relapsing disorder belonging to inflammatory bowel diseases (IBD), with characteristic skip lesions and transmural inflammation that may affect the entire gastrointestinal tract from the mouth to the anus. It is manifested by relapsing transmural inflammation found in any segment of the gastrointestinal tract. The disease may appear at any age, with the median age of onset being 30 years. CD is manifested by numerous uncharacteristic symptoms, but several stand out and constitute a typical presentation: chronic diarrhea and abdominal pain accompanied by weight loss, low-grade fevers, and fatigue. Despite the effectiveness of treatments (e.g., corticosteroids, immunosuppressants, and biological agents) for inducing long-term remission in adults with CD, secondary disorders, such as arthritis, osteoporosis, ocular inflammation, and skin lesions, as well as other extraintestinal symptoms, such as fatigue, depression, and anxiety, still frequently occur, resulting in a reduced quality of life. As it becomes increasingly clear that managing CD requires more than medical treatment alone, further research to identify second-line approaches for managing CD and its symptoms is necessary to address this public health concern.

Fatigue is a widely used term in both the medical literature and everyday clinical practice. However, it is relatively poorly defined and often subjectively interpreted. Although multiple definitions of fatigue can be found in the literature, there is a general agreement that in most cases, fatigue can be identified as a feeling of weakness, a sense of tiredness, a lack of energy, a feeling of exhaustion, reduced muscle strength, and cognitive impairment. In some patients, more atypical symptoms can be present [1].

Acute fatigue is a physiological condition experienced in everyday life as a predictable response to a prolonged period of physical exertion or stress. This short-lasting, transient feeling is relieved by rest and thus does not cause long-term impairment of function. Contrarily, chronic fatigue, which lasts for at least 6 months, and which cannot be cured by sleep or adequate rest, can be a sign of a somatic or psychiatric disorder [2,3]. In fact, fatigue is one of the most reported symptoms in primary care, with some studies showing a prevalence of up to 25% in the patient population [4], which is similar to the prevalence reported in newly diagnosed IBD patients [5]. However, in a 2012 online survey, more than 80% of 631 IBD patients who filled out the questionnaire (41% of which were in remission at that time) reported fatigue [6]. In a prospective, population-based IBD cohort study, fatigue was a symptom reported in 72% of patients with active disease and 30% of patients with inactive disease [7]. These findings suggest that although fatigue typically intensifies during periods of increased disease activity, it is also very prevalent in patients with clinical and endoscopic remission. What is also noticeable is that fatigue is more common among patients with newly diagnosed CD than in ulcerative colitis (48–62% for CD; 42–47% for UC) [8]. As reported by CD patients, chronic fatigue has not only a significant influence on the quality of life of patients, but it can also negatively affect the overall outcome of treatment. Due to the subjective nature of fatigue, in order to improve the quality of treatment and to individualize treatment, independent molecular factors that play a potential diagnostic role are sought. Current evidence on the efficacy of pharmacological CD therapy in the management of fatigue is limited, and some medications for the treatment of CD may even exacerbate fatigue [9]. This review summarizes the current literature on fatigue in CD and considers its etiology, diagnosis, and treatment.

## 2. Methodology

### 2.1. Searching Strategy

We carried out a systematic literature search to identify relevant original studies that enabled us to update the knowledge about chronic fatigue in CD patients and management of this disorder. The systematic literature search involved the following databases: OVID MEDLINE and EMBASE. The search query consisted of the combination of the following keywords: “Crohn’s disease”, “colitis”, ”gastrointestinal”, “chronic fatigue” and “myalgic encephalomyelitis.” Results were limited to papers relevant to the management of fatigue in CD patients that were published in English in 2013-2022. The first search was performed on 2 June 2022, and the search was updated on 15 August 2022, with a final revision on 10 December 2022. The selection of eligible papers is illustrated in Figure 1.

### 2.2. Study Selection and Risk of Bias

The references in all of the included studies were reviewed for more eligible articles. Each article was reviewed independently by six researchers (M.W., A.M., M.P., J.W., K.M., and J.F.) for inclusion according to the inclusion and exclusion criteria, which follow. Disagreements regarding article selection were resolved through discussion until a consensus was reached or the disagreement was resolved by discussion between authors M.W. and L.D. Prospective and retrospective observational human studies on adult patients were included. Conference abstracts were excluded. Articles were also excluded if they were not in English or if the studies were preclinical research or commentaries. A standardized form was used to extract data from the included studies. Extracted details were study population and demographics, details of interventions and controls, study methodology, and information to assess bias. Data extraction was performed independently by seven authors, and discrepancies were resolved through discussion with the other co-authors.

## 3. Etiology of Fatigue in CD Patients

Fatigue among CD patients can partly be explained by chronicity, disease activity, and nutritional deficits. However, the cause of CD-related fatigue currently remains unexplained in approximately half of patients, supporting the theory that fatigue can be an independent, systemic extraintestinal disease manifestation in IBD. An association between fatigue and clinically active IBDs has been known for a long time and is well explored in the literature. Recent studies have recognized several factors that may be associated with fatigue in CD patients.

## 4. Inflammation of the Gut

Multiple studies suggest the presence of a communication system between the gastrointestinal tract and the nervous system, the brain-gut axis. Fatigue in active IBD is hypothesized to be mediated by inflammatory cytokines and increased activity of T lymphocytes, primarily through the brain-gut axis [10]. High levels of experienced fatigue have been associated with higher levels of IL-10, IL-17A, IL-6, and interferon-γ (IFNγ), suggesting that inflammatory pathways play a role in fatigue pathogenesis [11]. In another study, the elevated levels of pro-inflammatory markers, such as tumor necrosis factor (TNF), IFNγ, and calprotectin, correlated with the severity of fatigue during active IBD [12]. On the other hand, one study showed that the levels of inflammatory cytokines do not differ between patients with and without fatigue during deep remission of IBD [13]. Inflammation is also associated with increased resting energy expenditure, which may cause increased fatigue. At the same time, pro-inflammatory cytokines can lead to anorexia and a decrease in caloric intake, dysregulation of the hypothalamic–pituitary–adrenal (HPA) axis, and the promotion anxiety and depressive symptoms by modulating the gut–brain axis [14,15]. Currently, the role of inflammation in the pathogenesis of fatigue in IBD is not fully understood, and further studies are warranted.

Gut microbial dysbiosis may induce fatigue through the brain–gut axis. It is well known that gut microbial dysbiosis has a significant influence on the propagation of inflammation in IBD. It is characterized by a decrease in valuable bacterial populations, such as *Faecalibacterium prausnitzii*, *Bacteroides fragilis*, and *Roseburia*, and an escalation of proinflammatory species, such as *Escherichia coli* [16]. Nagy-Szakal et al. supported this hypothesis with a study that involved two groups: patients with chronic fatigue syndrome (CFS) and healthy control individuals. Patients with CFS had decreased stool bacterial diversity [17]. Maes et al. reported that patients with CFS were characterized by increased levels of immunoglobulin (Ig) A and IgM. Higher levels of these Igs may be associated with an increase in lipopolysaccharide from enterobacteria. The intestinal epithelium should be a barrier against the translocation of lipopolysaccharide. This translocation may lead to the activation of innate immune responses [18]. Furthermore, in animal models, a reduction in depressive behaviors and anxiety was noticed in groups that received probiotics. A summarized overview of the included studies is presented in Table 1 [19].

## 5. Anemia

Anemia is an abnormal state associated with fatigue in IBD patients. It occurs in up to 20% of ambulatory patients and up to 68% of hospitalized patients with IBD [20]. A previous metanalysis, reported by Bartel et al., estimated that the prevalence of anemia among patients with CD is up to 27% [33]. Anemia may result from a wide spectrum of causes, which include malabsorption, impaired dietary intake, suppression of iron binding and erythropoiesis, chronic intestinal bleeding (visible or microscopic), inflammation, and certain types of medications, such as sulfasalazine, 5-aminosalicylates, and methotrexate [34].

The most common anemia in IBD is a result of iron deficiency, which is often caused by chronic gastrointestinal bleeding and decreased nutritional intake. Vitamin B12 and folate deficiency can also be linked to weakness and fatigue [35].

Due to the prevalence of anemia, patients with CD who complain about fatigue should be thoroughly investigated. Blood tests, such as tests for levels of red blood cells, hemoglobin, hematocrit, and levels of iron or vitamin B12, are basic, cheap tests that may be a valuable clue for every clinician.

Patients with established anemia, regardless of the cause, should be treated according to the cause of anemia (i.e., iron deficiency with parenteral or oral iron supplementation).

## 6. Psychological Factors

Despite rapid advancements in medical science in the last decades, CD remains so far incurable, though it can be treated with both pharmacological and surgical approaches. The primary aim of treatment is an achievement and maintenance of clinical and endoscopic remission for as long as possible. Unfortunately, relapses are common, and as such, many patients diagnosed with CD are presented with a prospect of a lifetime of recurrent absence from work and school due to sick leave and hospitalization, potential unemployment caused by the illness, health-related low quality of life, and even potential disability [21]. Such prospects, along with the troublesome nature of CD symptoms, may lead to an increased prevalence of experienced fatigue in patients with CD in comparison to healthy individuals [22]. It has even been hypothesized that the link between CD and chronic fatigue in patients during remission might be based mainly on an individual’s mental state, because a patient presented with a CD diagnosis is faced with challenges that for some might be too aggravating to endure on a daily basis. In a recent study, Radford et al. indicated that adults with IBD fatigue try to establish a sense of a ‘new’ normality by maintaining the same or a similar level of activities related to employment or education. However, this is often at the expense of personal, social, and leisure activities. Disease-related fatigue led patients to perceptions of conservation of energy through planning and prioritizing tasks. The authors also indicate that high levels of social support were associated with better self-reported health-related QoL in those patients with IBD, which indicates that underdiagnosed fatigue has a significant impact on impaired QoL [23].

Psychological factors, such as a depressive mood, stress, anxiety, and impaired QoL, are strongly associated with fatigue, and all of them are more common in CD patients than in the healthy population [36]. All these factors have been associated with increased fatigue scores as well as deterioration of the inflammatory disease course. Therefore, it is possible that a positive feedback loop exists whereby active disease leads to psychological distress that in turn aggravates the inflammatory state, with both factors consequently leading to increased fatigue. Additionally, abdominal pain, one of the main symptoms of active CD, has been shown to be associated with psychological distress, which in turn may affect sensory processing and thus lead to an increased perception of pain or the occurrence of chronic pain and/or fatigue. Screening for psychiatric disorders should thus be considered as an essential part of a holistic approach to fatigue in CD and justify a referral to an appropriate specialist.

A significant contributor to fatigue is sleep disturbance, which is prevalent in patients with CD with both active and non-active disease. Disrupted or restless sleep and multiple awakenings are reported to occur much more frequently in comparison to the general population [24]. A prospective study has shown that sleep disturbance increases the risk of CD relapse and is a significant factor in the worsening of patients’ life quality on par with previously mentioned psychological disorders.

## 7. Nutrient Deficiencies

Various nutrient deficiencies have been linked to fatigue in the general population [37]. IBD patients are at higher risk of nutrient deficiencies in comparison to the general population due to chronic inflammation, impaired muscle strength, and malabsorption that may be associated with the disease [38]. Restrictive diets, which are sometimes included in the therapy regimen, also often carry the risk of nutritional deficiencies unless properly fortified in lacking macro- or micronutrients. Patients with CD often harbor deficiencies of vitamins and minerals, such as vitamin B6 and B12, folate, ferritin, and zinc. This is observed mainly in patients with active disease and vitamin D deficiency in both active course and remission [39]. A list of vitamins and minerals linked with IBD is presented in Table 2.

Chronic fatigue has been proven to correlate with vitamin D deficiency in cancer patients, but these findings do not apply to CD patients. Some studies have showed that there is no direct association between fatigue in patients with IBD and vitamin D deficiency [25]. Nonetheless, nutrient status (iron, copper, zinc, folate, phosphate, magnesium, vitamin B6 and B12, calcium, and vitamin D) should be monitored and restored, with referral to a clinical dietician when appropriate and if necessary.

## 8. Screening for Fatigue in CD Patients

When fatigue is a persistent or especially pronounced symptom, a patient will generally report it to the physician. However, in many cases, it can be overlooked early and remain unrecognized. Routine screening for fatigue is an important initial step in clinical evaluation. This can be achieved by simply asking the patient if they feel or have recently felt fatigued.

There are also several screening tools that enable more thorough evaluation of fatigue. One of the quickest and easiest-to-use tools is the visual analogue scale (VAS) with a score from 0 to 10 covering the severity of fatigue, with 10 representing severe fatigue and 0 representing no fatigue [41] (Table 3).

This scale has been successfully applied in the evaluation of cancer-related fatigue to distinguish patients with mild fatigue (score of 0–3) from patients suffering from more severe fatigue (score of 4–10) [42].

Furthermore, there is the multidimensional fatigue inventory (MFI), a 20-item questionnaire, which measures fatigue in five dimensions: general, physical, motivation, activity, and mental. Similarly, the Multidimensional Assessment of Fatigue (MAF) scale has 16 items to measure fatigue in four dimensions, i.e., severity, distress, degree of interference with activities of daily living, and timing of fatigue. The Functional Assessment of Chronic Illness Therapy-Fatigue (FACIT-F) is a 13-question sub-scale of the Functional Assessment of Chronic Illness Therapy (FACIT) Measurement System (Table 4). FACIT-F enables the assessment of general fatigue but does not support the complex assessment of physical fatigue, mental fatigue, and activity, in contrast to MFI. The FACIT-F, however, has been validated to measure fatigue in chronic illnesses, such as IBDs, with good internal consistency, reproducibility, and sensitivity [26]. Unfortunately, there is a lack of consensus on which scale is best to use to measure fatigue in the IBD population. Of the mentioned scales, IBD-F is the only scale tailored specifically to patients with IBD.

## 9. Treatment of Fatigue in CD Patients

Before proceeding to specifically targeted interventions, general anti-fatigue strategies should be employed. In particular, teaching patients how to plan their days seems to be the crucial approach in anti-fatigue strategies. Patients should be advised to distribute their energy throughout the whole day and to plan for necessary rests and breaks. Furthermore, relatives play an important role in the process of acceptance of fatigue, as their acceptance and support are crucial in managing disease-related symptoms, thereby providing better therapeutic outcomes.

Non-pharmacological interventions, such as physical activity and psychosocial interventions, have been shown to help patients with a range of other chronic conditions to manage fatigue. Consequently, several non-pharmacological interventions have been applied in IBD populations, which are mainly focused on mental health symptoms or overall QoL. It has also been proven that the appropriate use of stress-management techniques has a beneficial effect on fatigue. There is evidence suggesting that electroacupuncture effectively reduces fatigue and increases QoL [27]. On the other hand, reduced activity and muscle strength are often reported in CD patients suffering from fatigue, and there is increasing evidence showing that physical activity is beneficial by improving bone health, increasing muscle mass and function, increasing energy intake, and possibly improving nutritional status; additionally, QoL and fatigue are improved by exercise in CD patients. Moreover, studies in animal models have suggested that exercise may reduce the inflammatory response [40,43].

Considering pharmacological interventions, there is no specific drug aimed at reducing feelings of fatigue alone. Various studies have considered commonly used drugs in CD and their influence on fatigue. Infliximab is an antibody against TNF. Patients, after administration of Infliximab, have often reported improvement in terms of fatigue [28]. Minderhoud et al. compared CD patients who received Infliximab with a placebo group. The placebo group initially reported a decrease in fatigue score, but at the conclusion of the study, they reported a recurrence of this condition. Meanwhile, the group that was administered Infliximab reported a decrease in fatigue, which continued to the end of the study. Despite the fact that the study was conducted on a small number of patients, Infliximab seems to be a possible treatment in both CD and severe fatigue [44]. Another drug, which is administered in patients suffering from CD, is Adalimumab. In a study reported in 2008, patients were divided into three groups: patients who received the drug only at induction, patients who received Adalimumab every week, and patients who received the drug every other week for 56 weeks. All groups reported a significant decrease in fatigue; however, the group that received the drug only once reported a recurrence of their symptoms after a few weeks [29]. Psychostimulants, such as methylphenidate and dexamethasone, have shown promising results in severe cancer-related fatigue [45]. However, these agents have not been investigated in IBD-related fatigue yet. Additionally, in a pilot study of 12 IBD patients with no preexisting thiamine deficiency, high-dose thiamine decreased overall fatigue scores [46]. In the study by Regev S et al., the short-term cognitive–behavioral and mindfulness intervention had the capacity to reduce chronic fatigue as well as improve functioning in patients with mild to moderate CD [30]. Nevertheless, randomized trials that have been performed in fatigued cancer patients have shown a significant placebo response. Consequently, there is no specific medical intervention that can reduce fatigue during remission, while at the time of relapse, underlying disease treatment according to the current guidelines should be initiated.

As mentioned before, multiple aspects should be taken into consideration during the development of fatigue therapy, i.e., anemia, nutritional deficiencies, mood disorders, sleep disorders, and some comorbidities [10]. Most of these are treatable; thus, they should be recognized, and appropriate treatment should be initiated in line with guidelines for the specific disorder.

## 10. Chronic Fatigue Syndrome and CD

In some cases, chronic fatigue can constitute a part of a larger cluster of symptoms called chronic fatigue syndrome (CFS) or myalgic encephalomyelitis (ME). CFS/ME is a severe multimodal disease with a high degree of physical disability, which leads to a high need for patient care. The disease is characterized by debilitating fatigue with unrefreshing sleep, neurocognitive impairments, and flu-like symptoms, such as muscle weakness and pain, headaches, sore throat, and tender lymph nodes. The malaise and the accompanying symptoms worsen dramatically after minimal physical, orthostatic, and cognitive activity. There is growing evidence that the gastrointestinal tract may play a role in the pathogenesis of CFS, but the exact link between these two disorders is yet to be determined.

A retrospective cohort study from 2019, which evaluated the risk of CFS in patients with IBD, showed that the incidences of CFS in women and men with IBD were 5.14 and 7.09 per 1000 person-years, respectively [31]. In the group of women and men without IBD, the incidences were 2.83 and 1.90 per 1000 person-years. Moreover, the incidence rates of CFS rose with age in both groups. Additionally, male sex was identified as increasing the risk of CFS. No difference in incidence rates was observed between patients with CD and other types of IBD. According to another study on IBD and CSF from 2018, higher scores of chronic fatigue were linked to clinically active disease in UC patients. However, this observation did not correlate with increased inflammatory markers [13].

Intestinal microbiota alterations and dysbiosis were detected in several CFS studies, but a specific consistent microbial signature was not found. There is an inconsistency in results, and thus an exact link between alterations in the intestinal bacteria and the disease mechanisms cannot be made. Newberry et al. [47] reported in their systematic literature review that there were eight agreeing and seven conflicting results between CFS microbiome studies, while there was overall evidence for dysbiosis.

Some researchers have proposed that bacterial translocation through the inflamed colon wall may be involved in the pathogenesis of CSF. This process is thought to be one of the main causes of CD as well, therefore suggesting that these two diseases may have a common pathogenesis. This idea is supported by the fact that serum IgA levels against the lipopolysaccharide (LPS) of enterobacteria are increased in patients suffering from CFS and are correlated with the severity of the disease. The elevated levels of bacterial components in the plasma of CFS patients are a result of increased gut permeability. According to Giloteaux and co-workers [32,48], it might be possible that increased bacterial proliferation in the gut results in high endotoxin levels and further damage of the epithelial barrier. The resulting infiltration of LPS into the bloodstream provokes the immune response and systemic inflammation. LPS also induces localized inflammation by binding to the toll-like receptor-4 complex. A mutation of Nucleotide binding oligomerization domain 2 (NOD2) leads to the binding of protein to the peptidoglycan of bacteria, which results in NF-κB activation and an inflammatory response; this may also play a role in the development of CD. Clinically, activation of NF-κB has been related to a feeling of tiredness [49].

Pro-inflammatory cytokines produced in the gut can then be transferred to the brain by the autonomic nervous system, causing an increase in cytokine levels in the brain and exacerbating neuroinflammatory processes, which are linked to the feeling of fatigue. For example, an increased level of IFNy is associated with fatigue and hyperalgesia [50].

## 11. Conclusions

Chronic fatigue is a common and underrecognized symptom of CD for which the prevalence is much higher in the population of CD patients compared to the healthy population. It stems from an intricate web of interactions between various risk factors, and its pathophysiology is still not fully understood. However, despite the wide range of available treatments, management of chronic fatigue remains a significant challenge for both CD patients and medical practitioners. Implementation of routine screening and a holistic, multidisciplinary approach involving psychological support may be crucial in the management of CD patients with chronic fatigue. There is currently no single intervention aimed at decreasing fatigue alone, and its treatment is especially difficult in patients with fatigue persisting despite clinical and endoscopic remission. The fatigue in CD is driven by various factors, and a multidisciplinary approach is crucial to manage fatigue. Further extensive research is still needed in order to be able to predict, prevent, identify, and ultimately treat fatigue associated with CD. It would be beneficial for future studies to examine different types of fatigue (physical, emotional, or mental fatigue). Finally, research is needed to develop effective fatigue interventions that can be easily translated into clinical practice.

## Figures and Tables

**Figure 1 life-13-01692-f001:**
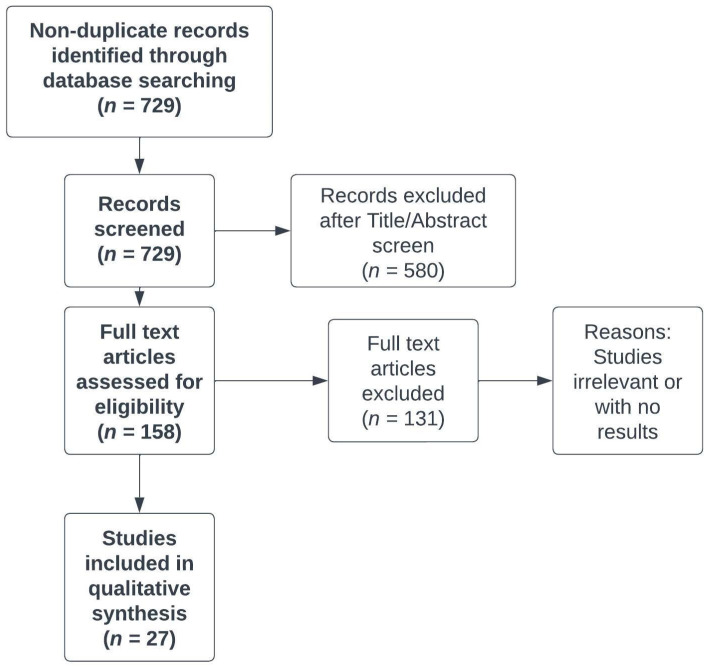
PRISMA flowchart of the systematic literature search and selection process. PRISMA, Preferred Reporting Items for Systematic Reviews and Meta-Analyses.

**Table 1 life-13-01692-t001:** A summary of human studies included in this review. (Abbreviations: CFS—chronic fatigue syndrome; COBMINDEX—CD-tailored cognitive–behavioral and mindfulness intervention; FACIT-F—Functional Assessment of Chronic Illness Therapy—Fatigue; IBD—inflammatory bowel diseases; IGF-1—insulin-like growth factor-1; QoL—quality of life.

Study	Region/Country	Year of Publication	IBD Patients’ Group Size	Outcomes
Cohen et al. [5]	USA, Rhode Island	2014	220 (125 patients with CD)	Fatigue is more prevalent in CD than in UC (30.6 vs. 22%) and is highly associated with poor QoL.
Danese et al. [6]	Europe, North America, Asia Pacific	2014	631	Patients with anemia and IBD report fatigue more frequently than individuals with IBD alone. If necessary, the supplementation of iron is recommended in an intravenous way.
Graff et al. [7]	Canada, Manitoba	2011	318 (160 patients with CD)	In total, 78% of patients with active CD develop high general fatigue.
Grimstad et al. [8]	Norway	2015	81 (20 patients with CD)	In total, 48–62% of newly diagnosed CD patients suffer from fatigue.
Lucia Casadonte et al. [11]	USA, Chicago	2018	67 children	Lower levels of serum IGF-1 are associated with more fatigue
Vogelaar et al. [12]	Netherlands	2017	55 (42 patients with CD)	Patients with fatigue are characterized by a significantly altered immune profile.
Jonefjäll et al. [13]	Sweden	2018	298	Psychological distress, iron deficiency, disease activity, and female gender are important factors for fatigue development.
Elsherif et al. [14]	UK, London	2014	494 (237 patients with CD)	Earlier age of diagnosis and ileitis are correlated with increased weight loss in CD patients.
Nagy-Szakal et al. [17]	USA	2017	50 non-IBD patients with CFS	CFS is characterized by gut dysbiosis.
Maes et al. [18]	Belgium	2007	29 non-IBD patients with CFS	Intestinal permeability is increased in patients with CFS.
Bager et al. [20]	Scandinavia	2011	437 (253 patients with CD)	Anemia was found in 23% of CD patients.
Jelsness-Jørgensen et al. [21]	Norway	2011	140 (48 patients with CD)	Chronic fatigue significantly reduces QoL in IBD patients.
Jelsness-Jørgensen et al. [22]	Norway	2011	140 (48 patients with CD)	Chronic fatigue is more prevalent among IBD patients than in healthy controls, but it cannot be predicted by epidemiological factors (with exception for smoking).
Radford et al. [23]	UK	2022	14 patients with CD	Fatigue negatively influences social activities of CD patients.
Hashash et al. [24]	USA	2022	232 patients with CD	Brief behavioral therapy for sleep in IBD and bupropion partially reduce fatigue in CD individuals.
Frigstad et al. [25]	Norway	2018	405 (227 patients with CD)	Fatigue is not associated with vitamin D deficiency in IBD patients.
Tinsley et al. [26]	USA, Massachusetts	2011	209 (132 patients with CD)	FACIT-F scale is a reliable and valid tool to measure fatigue in IBD.
Farrell et al. [27]	Systematic review with 14 included studies	2020	3741	Electroacupuncture and physical activity may reduce IBD-related fatigue.
Lichtenstein et al. [28]	USA, Pennsylvania	2002	105 patients with CD	CD patients treated with Infliximab reported significantly decreased fatigue.
Loftus et al. [29]	Canada	2008	778 patients with CD	Treatment with Adalimumab effectively reduces fatigue in CD patients.
Regev et al. [30]	Israel	2023	142 patients with CD	COBMINDEX significantly decreases fatigue observed in CD.
Tsai et al. [31]	Taiwan	2019	2163 (1991 patients with CD)	Male sex, age higher than 35 years, and absence of comorbidities are the factors increasing the risk of CFS development in IBD patients.
Giloteaux et al. [32]	USA, NY	2016	48 non-IBD patients with CFS	Intestinal permeability is increased in patients with CFS.

**Table 2 life-13-01692-t002:** Vitamins and minerals that are most often deficient in patients with IBD and complications of their deficiency [40].

Vitamin B6	Seborrhoeic dermatitis-like eruption, atrophic glossitis with ulceration, angular cheilitis, conjunctivitis, intertrigo, neurologic symptoms of somnolence, confusion, neuropathy (due to impaired sphingosine synthesis), and microcytic anemia
Vitamin B12	Fatigue, dizziness, breathlessness, headaches, mouth ulcers, upset stomach, decreased appetite, difficulty walking (staggering balance problems), muscle weakness, depression, poor memory, poor reflexes, confusion, pale skin, and paresthesia
Folate	Glossitis, diarrhea, depression, confusion, anemia, and fetal neural tube and brain defects
Ferritin	Hypothyroidism, fatigue, and dizziness
Zinc	Depressed growth, diarrhea, impotence and delayed sexual maturation, alopecia, eye and skin lesions, impaired appetite, altered cognition, impaired immune functions, and defects in carbohydrate utilization

**Table 3 life-13-01692-t003:** VAS numeric scale for fatigue.

VAS for Fatigue Questionnaire
How much fatigue are you feeling now?
1 2 3 4 5 6 7 8 9 10

**Table 4 life-13-01692-t004:** Mapping FACIT-Fatigue items to fatigue-related subcomponents identified during concept elicitation.

	Symptoms Concept				Impact Concept		
FACIT-Fatigue item	Level of sleeplessness	Lack of energy	Weakness	Lack of focus	Inability to maintain family and professional relationships	Decreased physical functioning	Frustration
1. Feeling Fatigue							
2. I feel weak							
3. I feel listles							
4. I feel tired							
5. I have trouble starting things							
6. I have trouble finishing things							
7. I have energy							
8. I am able to do my usual activities							
9. I need to sleep							
10. I am too tired to eat							
11. I need help in my everyday activities							
12. I am frustrated with being tired							
13. I have to limit my social activities because of fatigue							

## Data Availability

The datasets used and/or analyzed within the framework of this study are available from the corresponding author upon reasonable request.

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
