# Peer review of "The Role of Chronic Fatigue in Patients with Crohn’s Disease"

_life, 2023, doi:10.3390/life13081692_

Round 1

Reviewer 1 Report (Previous Reviewer 1)

In my opinion, the manuscript brings a comprehensive review of fatigue in patients with Crohn's disease. 

As a researcher and clinical physician, I consider the text will help the management of my patients and also as a warning for conditions that underlie this symptom. 

Reviewer 2 Report (New Reviewer)

Very interesting topic schould be published 

none

This manuscript is a resubmission of an earlier submission. The following is a list of the peer review reports and author responses from that submission.

Round 1

Reviewer 1 Report

According to the PRISMA CHECKLIST (http://www.prisma-statement.org/documents/PRISMA_2020_checklist.pdf) the authors should have provided an explicit statement of the objective(s) or question(s) the review addresses. In this context, the paper moved away from a more scholarly approach. The authors opted to present a practical and comprehensive of different aspects around fatigue. The text is relevant to the management of Crohn's disease patients and should be published. 

The research has addressed as main question chronic fatigue as a frequent clinical manifestation in patients with Crohn's disease. Chronic fatigue is a  relevant topic to be discussed in Crohn's disease patients as a target to be reached according to the International Organization for Study in IBD and it is barely discussed in the scientific literature: Turner D, Ricciuto A, Lewis A, D'Amico F, Dhaliwal J, Griffiths AM, Bettenworth D, Sandborn WJ, Sands BE, Reinisch W, Schölmerich J, Bemelman W, Danese S, Mary JY, Rubin D, Colombel JF, Peyrin-Biroulet L, Dotan I, Abreu MT, Dignass A; International Organization for the Study of IBD. STRIDE-II: An Update on the Selecting Therapeutic Targets in Inflammatory Bowel Disease (STRIDE) Initiative of the International Organization for the Study of IBD (IOIBD): Determining Therapeutic Goals for Treat-to-Target strategies in IBD. Gastroenterology. 2021 Apr;160(5):1570-1583. doi: 10.1053/j.gastro.2020.12.031. Epub 2021 Feb 19. PMID: 33359090. The text is a comprehensive review with a wide scope of fatigue etiologies and management. In my point of view systematic reviews are more appropriate for scientific publications, but a scoping review in this submission is very well composed and it is welcome. 

Reviewer 2 Report

Thank you for the opportunity to peer review this manuscript. I will not be recommending publication in this journal as I there are some aspects that are unclear in the manuscript and I also do not quite understand how studies were chosen. Please take my suggestions on board for future submissions to other journals.

Your stated aim at the end of the abstract needs to be clearer and align with the aim you state at the end of the introduction; they are slightly different.

You have done a systematic review seemingly but it lacks a table summarising all the studies that were accepted in terms of which outcomes they looked at, what their sample size was, and which design they were.

Your review is broad and reads like a narrative review even though you had a systematic review description in your methods.

At one point you say CD is subjective. I disagree. It is very objectively measured with clear physical criteria. What I think you are trying to say is that the patient's subjective experience of CD varies from person to person.

Why did you restrict your search to 2013 onwards? What were your specific reasons for excluding studies that you did exclude? Did you include systematic reviews and meta analyses?

If so, why did you not include D'Silva et al's meta-analysis despite its relevance in terms of diagnosis and risk factors (aetiology). If not, why did you cite Newberry et als systematic review? Apply criteria consistently.

You should have a Table summarising all included studies in terms of design, sample size, outcomes/what it reported, and any other relevant information such as which country it came from.

You need a smaller results section reporting just the results of the studies and a larger conclusion reporting your interpretations.

At the start of the 9. Treatment of Fatigue in CD patients you make a couple of specific claims in the first paragraph without citing them. If you are not citing them they are your own interpretation and belong in conclusion. If you are not giving your own interpretation, then you need to cite them. I am alluding to claims like "plan their days" and "distribute their energy"

For an example of how best to set out a systematic review, see this article: Waddell O, McCombie A, Frizelle F. Colostomy and quality of life after spinal cord injury: systematic review. BJS Open. 2020 Aug 27;4(6):1054–61. doi: 10.1002/bjs5.50339. Epub ahead of print. PMID: 32852897; PMCID: PMC7709367.

Reviewer 3 Report

This manuscript The role of chronic fatigue in patients with Crohn’s disease (CD) is a review article which is aiming at updating knowledge of chronic fatigue in CD patients and management in this disorder. "Chronic fatigue is a common and underrecognized symptom of CD, which prevalence is much higher in the population of CD patients as compared to healthy population."

Also, definition of this word is vague because different stand points are related and various risk factors interact, and its pathophysiology is still not fully understood. This review depicted most reliable mechanisms how fatigue persist during disease course.

 Author concluded “There is currently no single intervention aimed at decreasing fatigue, and its treatment is especially difficult in patients with fatigue persisting. Extensive research is still needed in order to be able to predict, prevent and ultimately treat fatigue associated with CD.”

I completely agree Authors opinion.